# Factors Affecting Incurred Pesticide Extraction in Cereals

**DOI:** 10.3390/molecules28155774

**Published:** 2023-07-31

**Authors:** Xiu Yuan, Chang Jo Kim, Won Tae Jeong, Kee Sung Kyung, Hyun Ho Noh

**Affiliations:** 1Residual Agrochemical Assessment Division, Department of Agro-Food Safety and Crop Protection, National Institute of Agricultural Sciences, Wanju 55365, Republic of Korea; yx0219@korea.kr (X.Y.); rlackdwh1@gmail.com (C.J.K.); shewaspretty@korea.kr (W.T.J.); 2Department of Environmental and Biological Chemistry, College of Agriculture, Life and Environment Science, Chungbuk National University, Cheongju 28644, Republic of Korea

**Keywords:** cereal, incurred pesticide, liquid chromatography–tandem mass spectrometry, milling, QuEChERS

## Abstract

This study investigated the effect of milling on the yields of incurred residues extracted from cereals. Rice, wheat, barley, and oat were soaked in nine pesticides (acetamiprid, azoxystrobin, imidacloprid, ferimzone, etofenprox, tebufenozide, clothianidin, hexaconazole, and indoxacarb), dried, milled, and passed through sieves of various sizes. The quick, easy, cheap, effective, rugged, and safe method and liquid chromatography–tandem mass spectrometry extracted and quantified the incurred pesticides, respectively. For rice and oat, the yields were higher for vortexed samples than for soaked samples. For rice, the yields improved as the extraction time increased from 1 to 5 min. The optimized method was validated based on the selectivity, limit of quantitation, linearity, accuracy, precision, and the matrix effect. For rice and barley, the average yields improved as the particle size decreased from <10 mesh to >60 mesh. For 40–60-mesh wheat and oat, all pesticides (except tebufenozide in oat) had the highest yields. For cereals, 0.5 min vortexing, 5 min extraction, and >40-mesh particle size should be used to optimize incurred pesticide extraction.

## 1. Introduction

Cereal crops constitute major staple foods worldwide and are regularly treated with numerous pesticides during the growing season and the post-harvest period [1,2]. Approximately 32% of all cereal yield may be lost if pesticides are not used. Therefore, it is essential to apply pesticides to ensure high crop yield [3]. Nevertheless, pesticide application leaves toxic residues and contaminates food crops. For this reason, appropriate analytical methods are required to measure and manage the levels of pesticide residues in foods such as cereals, fruits, and vegetables [4,5,6,7]. Recently, many chromatographic techniques have been used for the quantification of trace levels, such as liquid chromatography–tandem mass spectrometry (LC-MS/MS), gas chromatography–tandem mass spectrometry (GC-MS/MS), liquid chromatography–hybrid quadruple time of flight tandem mass spectrometry (LC-QqTOF-MS), and nano-liquid chromatography–high-resolution orbitrap mass spectrometry [8,9].

Previous studies analyzed pesticide residues in barley, oat, wheat [10], and rice [11]. The techniques used for this purpose included liquid–liquid partition [12], solid-phase extraction [13], and quick, easy, cheap, effective, rugged, and safe (QuEChERS) [14]. The QuEChERS method was first published in 2003 and has been routinely applied worldwide since then [15]. The widespread use of QuEChERS is due to its wide spectrum. QuEChERS can cover a wide range of pesticides, including polar, non-polar, acidic, and basic pesticides [16,17]. Moreover, the advent of QuEChERS has enabled easy analysis in a short period of time. Previous studies on the QuEChERS-based analytical method in various matrices, such as vegetables [18], fruits [19], soil [20], biological samples [21], and spices [22], have been published. However, even though the results of numerous QuEChERS-based pesticide analyses have been published, few of these have focused on the extractability of incurred pesticide residues in cereals [23]. It is difficult to use the QuEChERS method for extraction from cereals to analyze incurred pesticide residues because they may be trapped deep within the cereal particles [2]. Currently, the analytical method is usually validated with analyte spiking, but this approach does not reflect incurred residue levels [24]. Therefore, appropriate analytical methods, which can analyze incurred pesticide residues efficiently, are required [14].

Several previous studies optimized the QuEChERS-based method to improve pesticide extraction yield. To this end, they modulated various factors, such as extraction solvent acidity, sample moisture content, partitioning salts, and clean-up sorbents [19,20,25]. Here, however, we considered other factors, such as post-milling cereal particle size, water soaking time, and extraction time, because all of these can influence the extraction yield of incurred pesticides in cereal crops.

To the best of our knowledge, very few previous studies have compared the extraction yields of incurred pesticides in cereals by using different milled cereal particle sizes. Herrmann et al. reported that the extraction yield of eight pesticides in wheat improved by 31% when the sieve size was reduced from 5.0 to 0.2 mm [2]. In contrast, Hepperle et al. reported that the extraction yield of six different pesticides did not differ among milled and re-milled wheat samples [23]. Since these studies reported contrasting results for the same cereals, further research is required to draw more reliable conclusions regarding incurred pesticide extraction from milled cereals.

The method for analyzing pesticide residues in cereal matrices differs from that used in fruit and vegetable matrices, as only the former involves water soaking [23]; cereal samples are usually soaked in water for 20–30 min prior to analysis [25,26]. An earlier study optimized the soaking time required for incurred pesticide analysis [27]. However, pre-analysis cereal sample vortexing has seldom been considered. Therefore, the present study compared cereal sample soaking and vortexing.

Recently, several studies performed 1 min extractions to assess various physicochemical properties of pesticides [28,29,30,31]. Nevertheless, we hypothesized that a 1 min extraction would not fully remove incurred pesticides from the cereal samples. Moreover, very few studies have compared the yields of incurred pesticides removed from cereal samples at different extraction times. Hepperle et al. showed that the incurred pesticide extraction yield from raisins improved with extraction time [23]. In the present study, we postulated that extraction time impacts the yields of incurred pesticides extracted from cereal samples.

The objective of this study was to investigate the yield of incurred pesticides extracted from different rice, wheat, barley, and oat particle sizes. To determine the extraction efficiency, we identified the optimal soaking and extraction times of cereals and applied them in the comparative analyses of incurred pesticides extracted from cereal samples of different sizes.

## 2. Results and Discussion

### 2.1. Sample Processing

We immersed wheat samples (n = 3) for 30 min and 24 h and compared their incurred pesticide content. Appendix A shows that after 24 h, the average extracted pesticide residue content in the wheat was 1.5–2 times higher than that measured after 30 min of immersion. Therefore, 24 h was selected as the amount of time to immerse the cereal in the pesticides. The dried cereals were milled with high-speed laboratory knife mixers and passed through sieves of different sizes. Figure 1 shows the particle size distributions. The weight of the 10-mesh sample was approximately 15% of the total 2 kg of each cereal sample. The weights of the 10–20- and 20–40-mesh samples accounted for approximately 37% and 23% of the total 2 kg of each cereal sample, respectively. The weights of the 40–60- and >60-mesh samples and the lost material combined constituted approximately 25% of the total 2 kg of each cereal sample. The >40-mesh samples were produced by repeatedly milling the 20–40-mesh samples.

### 2.2. Effect of Soaking Time on Extraction Yield

The soaking time was optimized to reflect the extraction efficiency of each incurred pesticide. The soaking times required for dry commodities differ from those required for fruits and vegetables [23]. Previous studies showed that water addition and soaking time affect pesticide recovery from dry products, such as tea, Chinese medicine, and herbs [17,27,32]. However, these studies did not consider incurred pesticides. Moreover, a vortexing step prior to extraction was often omitted. However, brief vortexing can substantially reduce overall sample preparation time. We compared incurred pesticide extraction yields for rice and oat (40–60 mesh) vortexed for 0.5 min and soaked for 30 min (n = 3; relative standard deviation (RSD) ≤ 13.16). The average extraction yields of all compounds in all rice and wheat samples were higher after 0.5 min of vortexing than they were after 30 min of soaking alone (Figure 2). There were also different vortexing and soaking durations, which we could have compared. However, we considered the EURL recommended time for dry products and focused on reducing analysis time compared to the conventional method [33]. Based on the results, vortexing for 0.5 min not only ensured extraction efficiency but also reduced the analysis time. Hence, longer soaking times do not necessarily result in higher incurred pesticides extraction yield in rice and wheat samples. Guo et al. reported that the rates of pesticide recovery from green teas were higher after 30 min of soaking time than they were after 60 min [34]. Abbas et al. reported that soaking for 10 min showed better extraction yield in real herb samples (chamomile and parsley) compared to soaking for 15 min [35]. Therefore, 0.5 min of vortexing was used henceforth.

### 2.3. Effects of Extraction Time on Extraction Yield

Since extraction time is known to affect pesticide recovery [36,37], previous studies optimized extraction time to evaluate extraction yield [38,39]. In this study, the concentrations of all pesticides were highest in rice. Thus, incurred rice samples (40–60 mesh) were selected as a representative sample to optimize extraction time. Extraction time affects pesticide recovery [27,28]. Previous studies optimized extraction time to evaluate extraction yield [29,30,31,32]. Extraction times (1300 rpm for 1, 5, and 15 min) were compared (n = 3), and there was no significant difference between 1000 and 1300 rpm in terms of extraction yield. However, extraction yield increased as extraction time progressed from 1 to 5 min. On the other hand, the extraction yield did not markedly change between 5 min and 15 min extraction times. The hexaconazole and indoxacarb extraction yields were the lowest at 15 min of extraction time. Based on the foregoing results, the optimal extraction time was determined to be 5 min (Figure 3). Similar results were reported in a previous study; Wang et al. (2022) showed that the bifenthrin yield from strawberries improved significantly when the extraction time increased from 1 to 5 min. As the extraction time increases, the solvent can penetrate the waxy layers and can more effectively interact with the matrix [37]. Currently, a 1 min extraction time is common in the pesticide residue field. However, based on our results, it may take at least 5 min to ensure an adequate extraction yield of incurred residues.

### 2.4. Method Validation

Selectivity, limit of quantitation (LOQ), linearity, accuracy, and precision were validated to confirm the reliability of the experimental procedure (Appendix A). Selectivity was evaluated by analyzing blank rice, wheat, barley, and oat extracts. No interference was detected at the same analyte retention times. All pesticides in all cereals had linearity > 0.990. The LOQ for all pesticides in all cereals was set to 10 ng/g. The S/N ratios were >10, and the accuracy and precision of the measurements of all pesticides at concentrations of 10 and 50 ng/g in all cereals also met the aforementioned recovery criteria (70–120% and RSD ≤ ±20%). For the matrix effect, most of the pesticides showed minimum matrix effect in barley and medium matrix effect (−50% to −20%) in rice. All the pesticides showed signal suppression in all cereals. Cereals contain lipid components, and high-log Pow hydrophobic pesticides, such as etofenprox, may be susceptible to matrix effects. Appendix A shows the total ion chromatogram and combined spectra of nine pesticides. The analytical method used herein was validated before its application on samples of various particle sizes.

### 2.5. Effect of Particle Size on Extraction Yield

After method validation, we compared the incurred pesticide extraction yields for various sizes of cereal particles (n = 3). As the pesticides were highly concentrated, 2000-fold dilutions were conducted using blank extracts prior to LC-MS/MS injection. In this way, saturation was avoided, and optimal chromatogram peaks could be obtained. The average pesticide extraction yields for rice and barley improved as the particle size decreased from <10 to >60 mesh (Figure 4 and Figure 5). The azoxystrobin, etofenprox, tebufenozide, and indoxacarb extraction yields significantly increased with the decreasing particle sizes of rice and barley. The log *K*_ow_ values for non-polar azoxystrobin, etofenprox, tebufenozide, and indoxacarb were 2.5, 6.9, 4.25, and 4.65, respectively. In contrast, the values for imidacloprid, acetamiprid, and clothianidin were only 0.57, 0.8, and 0.7, respectively. These differences may be due to the fact that the aforementioned non-polar pesticides partition relatively faster in acetonitrile extracts [40]. The average pesticide extraction yields for wheat improved as the particle size decreased from <10 to >40 mesh, and they were highest in 40–60-mesh wheat particles. For oat, the average azoxystrobin, etofenprox, tebufenozide, and indoxacarb extraction yields also improved as the particle size decreased from <10 to >40 mesh, and they were highest in 40–60-mesh particles. The average acetamiprid, imidacloprid, ferimzone, clothianidin, and hexaconazole extraction yields were highest in 20–40-mesh particles. Oats had 2–3-times greater lipid content compared to rice, wheat, and barley [41]. Thus, non-polar pesticides, such as azoxystrobin, etofenprox, tebufenozide, and indoxacarb, are easily absorbed in the lipid portion compared to polar pesticides and show a dramatic increase in extraction yields because the small particles offer a larger interaction environment with acetonitrile. The preceding data were subjected to one-way analysis of variance (ANOVA); *p* < 0.05) and Duncan’s tests in IBM SPSS Statistics 25 (IBM Corp., Armonk, NY, USA; Appendix A). One-way ANOVA is the simplest statistical procedure used to test for differences between more than two groups. This procedure is based on the comparison of the range of a subset of the sample means with a calculated least significant range. When the *p*-value is small enough, it can be concluded that there is a significant difference between groups, and a *p*-value less than 0.05 is commonly used as the cut-off value for statistical significance [42]. The average extraction yields of all pesticides from rice and barley improved when the particle size was reduced from <10 to >60 mesh (*p* < 0.05). The average pesticide extraction yields were highest in 40–60-mesh wheat particles (*p* < 0.05). Except for tebufenozide in oat, the highest pesticide extraction yields were obtained using 40–60-mesh wheat and oat samples (*p* < 0.05). Overall, the incurred pesticide extraction yields were optimal when using 40–60- or >60-mesh cereal particles (*p* < 0.05). However, it is difficult to generate 60-mesh particles for routine analysis using a high-speed laboratory knife. Therefore, satisfactory pesticide extraction yields may be obtained by simply using 40-mesh cereal particles.

## 3. Materials and Methods

### 3.1. Chemicals

Standard analytical solutions (1000 μg/mL) of azoxystrobin, imidacloprid, ferimzone, etofenprox, acetamiprid, tebufenozide, clothianidin, hexaconazole, and indoxacarb were purchased from Accustandard (New Haven, CT, USA). Imidacloprid 8% SC was purchased from Bayer Crop Science Ltd. (Seoul, Republic of Korea). Acetamiprid + indoxacarb 9 (4 + 5)% SC was purchased from Hanearl Science Ltd. (Gangwon-do, Republic of Korea). Etofenprox 20% EC was purchased from Gyung Nong (Gyeongsangbuk-do, Republic of Korea). Azoxystrobin + difenoconazole 28.7 (17.4 + 11.3)% SC was purchased from Syngenta (Seoul, Republic of Korea). Ferimzone + hexaconazole 25 (15 + 10)% SC and clothianidin + tebufenozide 9 (1 + 8)% SC were purchased from Dongbangagro Corp. (Seoul, Republic of Korea). Methanol (LC-MS grade) was purchased from Fisher Scientific (Dublin, Ireland); formic acid (LC-MS grade) was purchased from Fluka (Charlotte, NC, USA); and acetonitrile (HPLC grade) was purchased from Merck GmbH (Darmstadt, Germany). Purified water was generated using an automatic purification system (Autwomatic Plus GR; Wasserlab, Navarra, Spain). The European Standard (EN 15662) method extraction kit (1 g sodium chloride, 4 g magnesium sulfate, 1 g sodium citrate, and 0.5 g disodium citrate sesquihydrate) and dispersive solid-phase extraction (d-SPE) (150 mg MgSO_4_, primary secondary amine (PSA) 25 mg) were purchased from Agilent Technologies (Santa Clara, CA, USA). Organic rice, wheat, barley, and oat were purchased from Coupang (Seoul, Republic of Korea). An Artlon gold mixer was purchased from Daesung Artlon Co. Ltd. (Paju-si, Republic of Korea).

### 3.2. Stock Solution Mixtures and Matrix-Matched Standard Solutions

For this step, 100 μL of each standard analytical solution (1000 μg/mL) of pesticide was transferred to a volumetric flask, and 9.1 mL of acetonitrile was added to produce stock solution mixtures. Stock solution mixtures (10 μg/mL) were prepared by combining the individual stock solutions with acetonitrile. The solutions used to plot the calibration curves were prepared in the 2–100 μg/L concentration range by serially diluting the stock solution mixtures with acetonitrile. The matrix-matched calibration curves were prepared in the 1–50 μg/L concentration range by combining the calibration curve solutions and each blank sample extract in a 1:1 ratio. All stock and working solution mixtures were stored at −20 °C until analysis.

### 3.3. LC-MS/MS Analysis

The LC-MS/MS analysis was performed in an Exion LC^TM^ fitted with an AB Sciex Triple Quad^TM^ 5500 (Sciex, Redwood City, CA, USA). A Kinetex C18 analytical column (3 mm × 100 mm; particle size 2.6 μm) was used at 40 °C oven temperature and with two mobile phases consisting of (A) 0.1% formic acid plus 5 mM ammonium formate in water and (B) 0.1% formic acid plus 5 mM ammonium formate in methanol. The mobile phase flow rate was 0.2 mL/min. The gradient conditions were as follows: 5% of the initial mobile phase B held for 0.5 min, ramped to 95% for 4.5 min, and held at 95% for 5 min. B was thereafter decreased to 5% for 0.1 min and held for 4.9 min to reach mobile phase equilibrium. The multiple reaction monitoring (MRM) and electrospray ionization (ESI) positive modes were used for the sample analyses. The retention times of acetamiprid, azoxystrobin, imidacloprid, ferimzone, etofenprox, tebufenozide, clothianidin, hexaconazole, and indoxacarb were 7.00, 8.25, 6.80, 8.50, 10.64, 8.70, 6.80, 8.90, and 8.86, respectively. The precursor, quantification, and qualification ions are shown in Appendix A. The curtain, collision, nebulizer, and drying gas pressures were 35, 10, 50, and 50 psi, respectively. The ion source temperature was 550 °C, and the positive ion spray (IS) voltage was +5500 V. The data were processed with MultiQuant™ 3.0.2 (v. 3.0.8664.0; Sciex).

### 3.4. Sample Processing

Azoxystrobin, imidacloprid, ferimzone, etofenprox, acetamiprid, tebufenozide, clothianidin, hexaconazole, and indoxacarb mixtures were formulated at 87, 100, 75, 100, 100, 100, 12, 50, and 80 μg/mL in 2 L of water. These pesticides, which we selected, are registered for use in rice in Korea. Two-kilogram lots of rice, wheat, barley, and oat seeds were soaked in the pesticide solutions in five-liter plastic beakers for twenty-four hours to scale down the previous laboratory dipping test [43]. The samples were then air-dried under a laboratory hood for 5 d, milled with a high-speed laboratory knife mixer, and passed through 10-, 20-, 40-, 60-mesh sieves to obtain <10-, 10–20-, 20–40-, 40–60-, and >60-mesh cereal particles. The milled samples were placed in polyethylene zipper bags and stored at −18 ℃ until analysis.

### 3.5. Soaking Time Optimization

Incurred rice and oat samples (40–60 mesh) were used for soaking time optimization. The performance of 0.5 min of vortexing was compared against that of 30 min of soaking in water (n = 3). All other parameters were the same as those used for the sample preparation.

### 3.6. Extraction Time Optimization

Incurred rice samples (40–60 mesh) were used for extraction time optimization. The 1, 5, and 15 min extraction times at 1300 rpm were compared (n = 3). All other parameters were the same as those used for sample preparation.

### 3.7. Sample Preparation

For sample preparation, 5 g cereal samples were weighed out in a 50-mL Falcon centrifuge tube; 10 mL of distilled water was added, and the suspension was vortexed for 0.5 min. Analytes were extracted with 5 min of vigorous shaking in 10 mL of acetonitrile in a Geno/Grinder homogenizer (SPEX SamplePrep, Metuchen, NJ, USA). The extraction salt package was added, and the suspension was shaken vigorously for 1 min and centrifuged at 2800× *g* for 5 min in a Combi-514R centrifuge (Hanil Science Co. Ltd., Incheon, Republic of Korea). One milliliter of the upper layer was transferred to a d-SPE tube containing 150 mg of MgSO_4_ plus 25 mg of PSA, after which the mixture was vortexed and centrifuged at 12,000 rpm for 5 min. The upper layer was mixed with acetonitrile in a 1:1 ratio for the LC-MS/MS analysis.

### 3.8. Method Validation

The method was validated based on selectivity, LOQ, linearity, accuracy, precision, and the matrix effect. Blank rice, wheat, barley, and oat sample extracts were used to evaluate the selectivity. The LOQ was set to S/N ratio >10, and the concentrations used fulfilled the recovery and precision criteria, namely 70–120% and RSD ≤ ±20%. The matrix-matched calibration curve linearity was expressed in terms of the coefficient of determination (*R*^2^). The recovery test accuracy and precision at 10 and 50 ng/g (n = 3) were evaluated using the criteria in Rural Development Administration: 70–120% and RSD ≤ ±20%. The matrix effect of each compound was evaluated using the slope of the matrix-matched standard calibration curve and pure solvent standard calibration curve as follows: matrix effect (%) = (slope of the calibration curve obtained using matrix-matched solution/slope of the calibration curve obtained using pure solvent − 1) × 100 [19].

### 3.9. Statistical Analysis

One-way analysis of variance (ANOVA) and Duncan’s tests in IBM SPSS Statistics 25 were used to compare the difference of incurred pesticide content by particle sizes (*p*-value < 0.05).

## 4. Conclusions

This study investigated the effects of soaking time, extraction time, and particle size on the extraction yields of incurred pesticides in rice, wheat, barley, and oat. All the foregoing factors substantially influenced the incurred pesticide extraction yield. Samples which were vortexed for 0.5 min after 30 min of soaking resulted in higher extraction yields than samples which were soaked alone. Extraction for 5 min resulted in higher yields than extraction for 1 min. Based on our results, it may take at least 5 min to ensure the adequate extraction yield of incurred residues. After optimization of the soaking and extraction times, the method was validated based on selectivity, LOQ, linearity, accuracy, precision, and the matrix effect. The average incurred pesticide extraction yields for each cereal type and particle size were compared using the following optimized parameters. For rice and barley, the average extraction yields of all nine pesticides improved when the particle size was reduced from <10 to >60 mesh. The average pesticide extraction yields for wheat improved as the particle size decreased from <10 to >40 mesh, and they were highest in 40–60-mesh wheat particles. For oat, the average azoxystrobin, etofenprox, tebufenozide, and indoxacarb extraction yields also improved as the particle size decreased from <10 to >40 mesh, and they were highest in 40–60-mesh particles. One-way ANOVA showed that the average extraction yields of all pesticides from rice and barley improved when the particle size was reduced from <10 to >60 mesh. Except for tebufenozide in oat, the highest incurred pesticide extraction yields were obtained for 40–60-mesh wheat and oat samples (*p* < 0.05). Overall, the incurred pesticide extraction efficiency was optimal when the cereal particles were 40–60 or >60 mesh in size. As it is impractical to generate the latter, cereal particles >40 mesh in size should be used to achieve ideal incurred pesticide extraction efficiency in routine laboratory analysis.

## Figures and Tables

**Figure 1 molecules-28-05774-f001:**
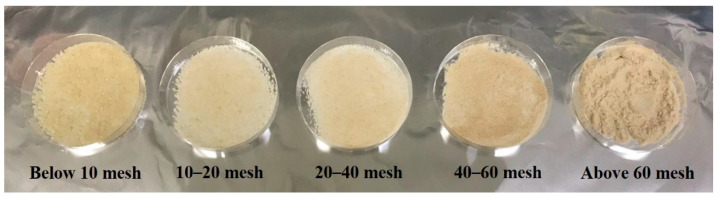
Rice samples classified by particle size.

**Figure 2 molecules-28-05774-f002:**
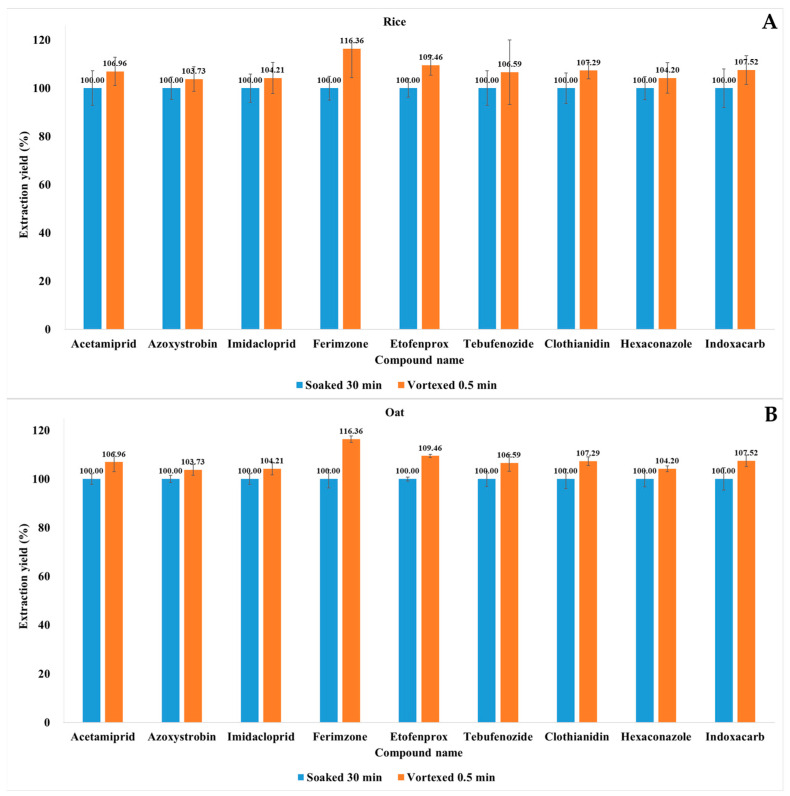
Effect of soaking time on incurred pesticide extraction yield. Average extracted pesticide residue concentrations in rice (**A**) and oat (**B**) soaked for 30 min were set as 100% extraction yield.

**Figure 3 molecules-28-05774-f003:**
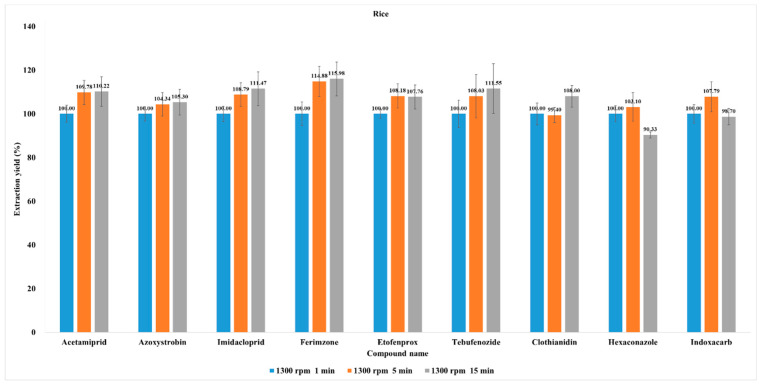
Effect of extraction time on incurred pesticide extraction yield. Average extracted pesticide residue concentration in rice spun at 1300 rpm for 1 min was set as 100% extraction yield.

**Figure 4 molecules-28-05774-f004:**
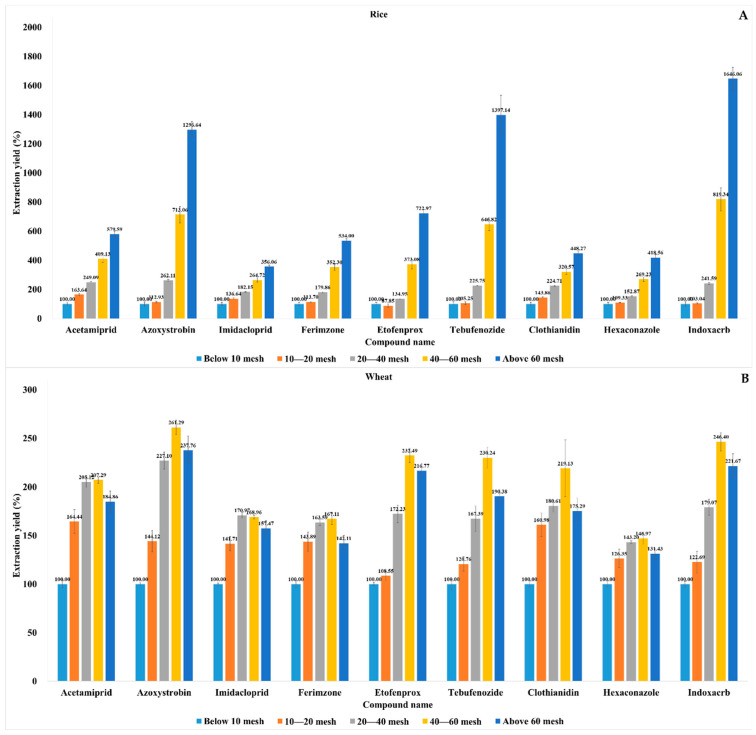
Effect of cereal particle size on incurred pesticide extraction yield. Average extracted pesticide residue concentrations in rice (**A**) and wheat (**B**); particles with <10 mesh were set as 100% extraction yield.

**Figure 5 molecules-28-05774-f005:**
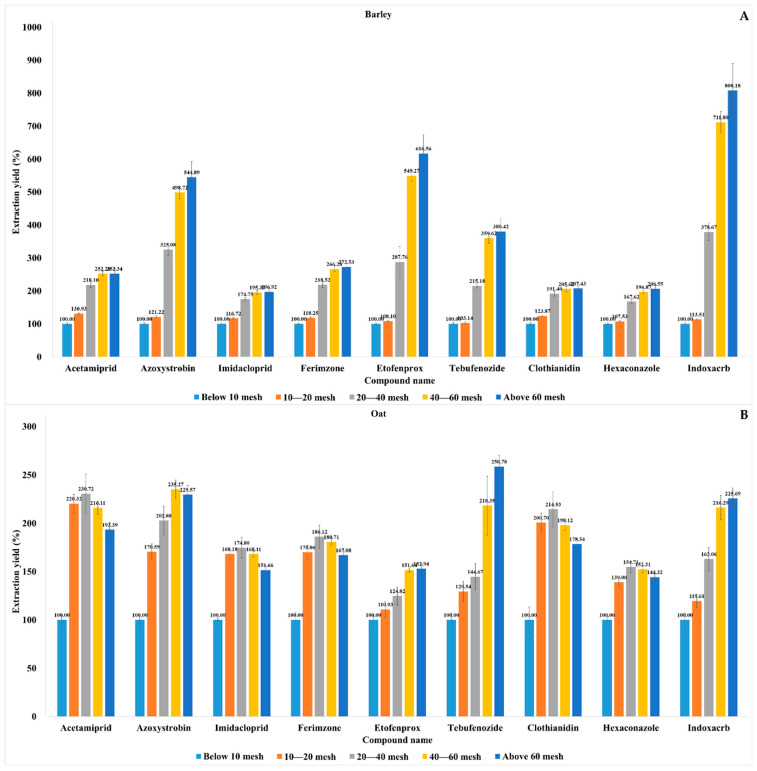
Effect of cereal particle size on incurred pesticide extraction yield. Average extracted pesticide residue concentrations in barley (**A**) and oat (**B**); particles with <10 mesh were set as 100% extraction yield.

## Data Availability

Data are contained within the article.

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
