# Peer review of "Factors Affecting Incurred Pesticide Extraction in Cereals"

_molecules, 2023, doi:10.3390/molecules28155774_

Round 1

Reviewer 1 Report

The development of new methods for food safety is ceartinly welcome. In this study, the authors investigate the yield of incurred pesticides extracted  from different samples, including rice, wheat, barley in order to determine extraction efficiency. The results are looking promising. However, there are several issues that the authors should address and fix to reach acceptance level in molecules as follows;

1) Manuscript English needs improvement.

2) page 2 line 55  ..."The method of analyzing pesticide residues in cereal matrices differs from that used on fruit and vegetable matrices as only the former involves water soaking..."  the authors should discuss recent and new LC-MS methodologies regarding pesticide  analysis in foods  (e.g. 10.1039/C8AY02173B).

3) Linear range should be more extensive (e.g. 0.1-500ppb)  

4) page 5 line 127

LC-MS/MS spectra of pesticide with TIC regrading pesticide used in the study should be given.

5) page 5 line 150

"The average imidacloprid, ferimzone, and hexaconazole extraction yields were similar for 10–20- mesh, 20–40-mesh, 40–60-mesh, and >60-mesh oat particles. Nevertheless, the average azoxystrobin, tebufenozide, and indoxacarb extraction yields were optimal for > 60-mesh oat particles. The preceding data were subjected to one-way ANOVA (p < 0.05) in IBM SPSS Statistics 25 (IBM Corp., Armonk, NY, USA) (Table S3)...."

Some MS spectra of findings  should be given to increase article quality.  

needs improvement

Reviewer 2 Report

The work presented for review is quite interesting. Prevention from exposure to pesticide residues is extremely important and I consider the work to be much needed.

Below are my comments. I hope the authors will consider including them in their revision as I strongly believe that these will improve the quality of the article.

Please complete the abstract with the number and types of tested pesticides and validation parameters. What was the spiking levels of samples?

There is also a lack of information on the effectiveness of the techniques used. Were post-operation pesticide residues below the MRL’s?

Figure 1 should present the fractions at higher magnification. Differences are barely visible in the current image. Maybe microscopic images would be better? It is known that the larger the contact surface of the solvent, the higher the efficiency of the process.

Section 2.2 The soaking effect was assisted by vortexing, thus comparison with only the soaking is not valuable. What with the soaking assisted by mixing? It should be emphasized that supporting this process enables shortening of the analysis time.

Lines 103-104 should be corrected.

Authors should compare different vortexing times and draw conclusions based on the results, rather than comparing results from other studies. Especially since only the soaking time was tested in them.

What with the other sample matrices? The results presented only rice and oat. 

In Figures 2-5 and in the Supplementary Materials please present the data with an accuracy of two significant figures. If the bars contain values, there is no need to keep the values on the Y-axis.

The extraction time was optimized only for rice samples. What with the others matrices? 

According to the statement in lines 121-122, why Authors did not test longer than 5 minutes?

The chromatogram for extract at LOQ level should be added in the Supplementary Material. The LOQ levels (lines 131-132) should be presented as mg/g. If standards were added for mass of solid matrix. If the concentration levels correspond to those of the spiked soaking water, the accumulation factors must be considered as each matrix has different characteristics.

Information on the study of the matrix effect is lacking, and due to the different crop samples, this effect may have a significant impact on the results.

In section 3.1 there is no explanation of the abbreviations used.

Does the description of standards used mean that they were in the form of a mixture at different concentration levels? If so, how were the standard mixtures prepared? Gravimetrically or volumetrically?

Were matrix-matched calibration curves prepared for each matrix separately?

What proportions of masses were obtained in relation to 2 kg of sample for different fractions after sieving? 

How to keep the representativeness of the sample taken for further analysis?

Why only 0.5 min and 30 min of soaking time were investigated? The longer the time, the greater the efficiency of the process.

Why extracts were diluted with acetonitrile before LC-MS/MS analysis? 

Round 2

Reviewer 1 Report

The authors addressed to all my concerns and the revised version of the manuscript is now suitable for publication in Molecules.

Reviewer 2 Report

I recommend for publication.
